# Contextualizing Risk Perception and Trust in the Community-Based Response to Ebola Virus Disease in Liberia

**DOI:** 10.3390/ijerph18063270

**Published:** 2021-03-22

**Authors:** S. Harris Ali, Kathryn Wells, Jarrett Robert Rose

**Affiliations:** Department of Sociology, Faculty of Liberal Arts and Professional Studies, Keele Campus, York University, Toronto, ON M3J 1P3, Canada; hali@yorku.ca (S.H.A.); jarrettr@yorku.ca (J.R.R.)

**Keywords:** Ebola virus disease, risk perception, trust, Liberia, community engagement, focus group discussion

## Abstract

The 2014–15 Ebola Virus Disease (EVD) outbreaks in Western Africa became widespread in primarily three countries, Guinea, Liberia, and Sierra Leone. Unlike all previous outbreaks in Central and East Africa, which were confined to rural areas, the virus spread rapidly through West Africa as a result of transmission through high-density urban centres coupled with the effects of public distrust in outbreak response teams and local government officials. *Objective:* In this study, we examine the EVD epidemic in Liberia, the first country to implement a community-based response that led to changes in the trajectory of the epidemic. The focus on the role of community-based initiatives in outbreak response is often neglected in conventional epidemiological accounts. In this light, we consider the manner in which community-based strategies enabled a more effective response based on the establishment of better trust relations and an enhanced understanding of the risks that EVD posed for the community. *Methodology:* We conducted qualitative research in five distinct communities in Liberia three years after the outbreaks subsided. Data collection procedures consisted of semi-structured interviews and focus group discussions with residents. *Results:* We found that the implementation of a community-based response, which included the participation of Ebola survivors and local leaders, helped curb and ultimately end the EVD epidemic in Liberia. As community members became more directly involved in the EVD response, the level of trust between citizens, local officials, and non-governmental organization response teams increased. In turn, this led to greater acceptance in abiding to safety protocols, greater receptiveness to risk information, and changes in mobility patterns—all of which played a significant role in turning the tide of the epidemic.

## 1. Introduction

Ebola Virus Disease (EVD) has historically been a disease with a fatality rate limited to rural locations within Central and Eastern Africa. In contrast, the 2014–2015 spread of EVD crossed the rural urban divide of West Africa [1] and led to a devasting and unprecedented loss of over 11,000 lives in Liberia, Sierra Leone and Guinea [2]. In 2014, EVD had spread to all three countries and citizens had limited access to medical facilities (see Figure 1). The dramatic increase in scale of the West African epidemic has been attributed to numerous features associated with these specific countries, including: inexperience in dealing with EVD, a lack of resources and physical infrastructures to combat the outbreaks (i.e., water, sewage, electricity, and road networks), high levels of mobility between communities, and weak health systems [3]. Such explanations, however, as Wilkinson and Leach [3] observe, are woefully de-contextualized and de-politicized and explanations that better capture the actual reality will require consideration of a different set of criteria, namely those that are explicitly rooted in the historical trajectory of the region (ibid).

In light of the criticisms raised by Wilkinson and Leach [3], if we do consider the broad political economic trajectory of the three countries, we see that they are generally comparable. As Howard [4] notes, they have very similar histories in the sense that they are all predicated upon the establishment of extractive economies and structural poverty, foreign intervention, colonial rule, and patrimonial regimes. In the case of Liberia and Sierra Leone further hardships were experienced because of the onset of recent civil wars. Notably, within this shared regional political economic context, government funding and emphasis were directed exclusively towards facilitating the private enrichment of foreign-owned resource companies and local elites. As a consequence, projects that would benefit the collective, such as social welfare programs, networked infrastructures (e.g., water, sewage, electricity, roads) and notably public health programs, were all subject to serious neglect by the state. In particular, Azétsop et al. [5] note that such state neglect facilitated the process of converting health into a privatized commodity as opposed to being viewed as a social good. This shift in conceptualization contributed to the total collapse of the public health systems in these countries while leaving the provision of health care to be dependent almost exclusively on the work of non-governmental organizations (NGOs). Furthermore, as Richards et al. [6] note, many rural communities did not have any experience with biomedical diagnosis and treatment. The resultant conditions made each nation-state seriously ill-prepared to face the Ebola crisis [4]. Nevertheless, despite facing formidable challenges, the three West African nations were able to eventually contain EVD spread. By the end of 2014 and beginning of 2015 the responses in each country began to mirror each other’s success [1]. Notably, the crucial link that contributed to their common success was an intervention based on some form of community-based involvement and the mobilization of locally trusted institutions [7]. Liberia was the first of the three countries to formally implement community-based initiatives in the EVD response and these initiatives played a key role in reversing this country’s steep epidemic trajectory several weeks before this was to occur in Sierra Leone and Guinea [8].

Community-based initiatives refer to strategies in which there is the direct involvement and active participation of community members in the various phases and aspects of the EVD response. Sharon Abramowitz and her colleagues [9] detail how such initiatives may be mounted during the prevention, response/treatment and aftermath phases of the outbreaks and the reader is referred to that work for a comprehensive overview (see also [5,7,10]). For the purposes of the present discussion, we can briefly say that the community-based initiative refers to a strategy in which community members are explicitly involved in important EVD prevention efforts such as: response training (including case identification and contact tracing), raising community awareness, hygiene, surveillance, the creation of local support infrastructures for the provision of food and water for those quarantined, and implementing isolation, quarantine and triage measures (ibid: [4]).

Abramowitz et al. [9] notes that community-based initiatives undertaken by communities themselves often remain unaccounted for in analyses of outbreak response. In this paper we hope to help address this deficiency in the literature by analyzing the lived experiences people had in relation to community-based interventions. Specifically, we focus on how the adoption of a community-based intervention may have prompted a change in people’s risk perceptions and the level of trust they invested in other groups as the EVD outbreaks unfolded in Liberia. In particular, as part of our focus, we consider how the initially unsuccessful bid to curb EVD spread compared to the successful response based on community-based initiatives.

## 2. Materials and Methods

This is a qualitative study based on five focus group discussions and key informant interviews conducted in five distinct communities situated within the Liberian counties of Montserrado, Lofa, and Margibi (see Figure 2). The data was collected by the Dr. Mosoka Fallah and his Community-Based Initiative research team based in Monrovia about three years after the end of the EVD pandemic. At each of the five sites we conducted focus groups consisting of ten participants, and ten key informant interviews, resulting in a total of 100 participants for our study, all of whom were over 18 years of age at the time of the research. All focus groups were conducted in late December 2018 except for the informal settlement community of West Point, Monrovia which was done in October 2019. The focus group discussions lasted from about 60 to 90 min while interviews lasted between 30 to 60 min.

Recruitment was done by first meeting with community leaders to discuss the aims and objectives of the study and to seek their permission for conducting interviews with community members. We were then directed to members within the community who were: EVD survivors, individuals who had acted as caregivers of Ebola cases during the outbreak, family members of deceased cases, response workers during the Ebola outbreak and community leaders who had themselves been involved in the EVD response. Voluntary consent was obtained from all participants, all of whom were compensated for their time with US $5.00. To ensure confidentiality, participant names were not recorded, and the interviews were conducted in semi-private places. The focus group participants were each provided a coded number to avoid communication of names, again to help ensure confidentiality. A portable voice recorder was used to record the proceedings.

Where necessary, transcribed material was translated from the local vernacular to standard English. Liberian Kreyol English was used at all sites except Foya in Lofa County where an interpreter helped translate questions to Kissi for participants. The transcripts were systematically coded manually using a thematic analysis approach. The most salient and common themes regarding risk perceptions and trust during and after the EVD outbreaks were noted and served as the basis of our analyses. Such themes that emerged from the data include isolation and fear, behavioral changes, migration or mobility, restrictions and response, misinformation, community engagement, and safety measures. Here we focus on risk perception, behavioural changes, mobility, trust and distrust.

This research was part of a larger study whose overall objective was to identify the lessons learned from adopting a community-based approach in Liberia (and Sierra Leone) during the 2014–2015 outbreaks for the purposes of training officials in the Democratic Republic of the Congo in this approach as they engaged in response efforts during the 2018–2020 EVD epidemic. Thus, the interview and focus group questions were geared towards that objective and included questions such as: “Tell us about how Ebola affected your community, what changes did it cause during the outbreak and after the outbreak?” What aspects concerning the Ebola response made you feel bad and why?” What in your opinion was good about the Ebola response?” “In what way did you resist or facilitate the response?”

The research protocol was approved by the Human Participants Review Sub-Committee of the Office of Research Ethics at York University and the National Research Ethics Board of Liberia (JFK Medical Center, Monrovia, Liberia).

The five communities selected for our study were chosen because they were identified as sites that had experienced clusters of EVD. The community of Foya in Lofa County is situated about 70 km south-west of Meliandou, Guinea—considered to be the index site of the West African EVD outbreak [11]. Foya is situated close to the boundary with both Sierra Leone and Guinea and the inhabitants of this community are primarily from the Kissi, Lorma, Gbande, and Mandingo tribes. The communities of Banjor, Red Hill, Nyanford Town, and West Point are located in Montserrado County and are all within or near the vicinity of Monrovia. These communities include inhabitants from tribes originating from all 15 counties of Liberia. The Needowein community is located in Margibi County (situated south west of Monrovia in the adjacent county) and consists of people primarily from the Bassa and Kpelleh tribes. The livelihood of people in the five communities ranges from agriculture and business for residents of Foya and Nyanford Town, fishing activity and petty trade in West Point and Banjor, farming, charcoal burning, and fishing for residents of Needowein, and fishing and petty trade in Banjor.

For our analyses, we make a rough distinction between the earlier and later phases of the response in order to gain an idea of how changes in the social context of the response were associated with changes in the risk perception and social trust over the course of the outbreak. One important proviso to our approach is that the qualitative nature of our data establishes certain constraints on precision in several ways. First, we do not use a specific reference date to distinguish the earlier from the later phases although analytically we understand the tacit cut-off to be the time that community-based initiatives started to be rolled out in the communities—the specific times that this happened varied from community to community. We adopt such an approach for pragmatic reasons. For most study participants a precise and clear-cut distinction was not part of their thinking and it would be artificial and awkward (not to say confusing) to impose a specific date during our interviews and discussions. Temporality however can be ascertained or inferred based on understanding the verbal context and flow in which the peoples’ responses were given. Second, since our study is retrospective and cross-sectional, we necessarily run into the problem of recall bias. Third, our study is based not on probability sampling but purposive or judgmental sampling. For our purposive study, the limitations faced in these regards are less crucial than for studies based on conventional epidemiological study designs whose aims are defined by the need for statistical inference to the population. Our objectives are more modest. Our main objective is to gain qualitative and theory-informed insights into the socio-political and subjective dimensions of the outbreak response—especially as they relate to community-based interventions and the impact they have had on risk perception and trust. Thus, although we acknowledge the issues pertaining to imprecision, recall bias, and non-probability samples, such limitations do not invalidate our qualitative approach to capturing insights into the subjective impressions people had of the community-based initiative as part of the overall EVD response.

The concepts of “risk perception” and “trust” we deploy in the analysis here have a long and well-developed history in the broader academic field of risk analysis—see for example the work of prominent theorists Ulrich Beck [12,13] and Anthony Giddens [14,15,16]. We develop these concepts in a more precise and fuller fashion in a theoretically-informed companion piece to this article (Anonymous, forthcoming) where we develop the overarching perspective of the “riskscape” as a way to analytically capture the concepts of risk perception and trust within a more cogent theoretical framework. For the purposes of the current, empirically-based analysis, we conceptualize these notions in a less nuanced but pragmatic manner. Risk perception we define simply as the level of conceived threat people associate with EVD. We deploy the concept of trust to refer to the level of faith people invest in others (or institutions) with respect to the truthfulness and reliability of the claims these others make [17,18].

## 3. Background

The colonial history of resource extraction in Liberia contributed to the perception people had of outsiders. As Azétsop et al. [5] note, resource extraction has been pursued at the expense of local populations, most vividly seen in the case of expropriation of land from the local peasantry by mining companies in conjunction with local government powers. Such situations understandably led to resentment of both foreign interlopers as well as government authorities, even when such parties were involved in ostensibly unrelated matters such as the EVD response. This corroborates the claim of Batty [19] and McGovern [20] that the long-standing hostilities between local people and the political and economic elites created a climate of suspicion towards any state-sponsored attempts to intervene in local affairs.

The effects of the history of corrupt practices were not limited to the business sector. As alluded to previously, Leach [21] notes that the funneling of state funds for exclusively private sector development such as mining and large-scale agriculture led to a chronic underinvestment in the health sector. This not only left the health system technically weak but contributed to the overall disintegration of basic health system capacity altogether. In other words, the decimation of the public health system in Liberia was the result of corruption that reduced the number of resources available to the health sector and health systems. In Liberia there were only 10,052 health care workers (mostly located in Monrovia) and many highly trained health staff such as doctors, nurses and other health professionals fled the country for safety during the civil war period [22]. Further, there was an absence of a disease surveillance systems and no inter-jurisdictional or inter-agency protocols in place for managing epidemics [22,23]. Thus, as Leach [21] observes, the deep under-investment in health services prior to Ebola meant that there were no nurses, doctors, and drugs available in the formal health settings. Consequently, the local population had limited experience with health care systems [6] and realized that the arduous trip to NGO or government run health centre was useless so they turned to other, alternative health providers as the basic source for health provision. In this context, the idea of going to a health centre and not caring for people at home was seen as alien.

Reluctance to travel to health centres was reinforced by local knowledge of the corrupt practices of government elites, which weakened public trust invested in what remained of the healthcare system. The lack of trust in international (NGO) actors and the legacy of colonial politics meant that the distrust in state health services was rooted in a broader distrust of state officials and foreigners more generally [21]. People could see the difference in attention being paid to Ebola response, by governments and international actors, in comparison to other diseases such as malaria [6]. This difference was, to some, proof of a hidden agenda, which affirmed misinformation about the disease. People also understood the commonalities in the shared ideology and approach of the state and foreign elite with respect to the exploitation of labour, resources and slavery at the expense of building up local livelihoods. People saw evidence of this during the early stages of the EVD response with respect to how foreigners treated them poorly, in terms of disrespectful and impolite interactions, especially in the treatment of the deceased bodies [21]. Based on past experiences, people’s cultural rationality was informed by the idea that outsiders did not hold their best interests at heart. It was in this context, with these types of understandings that various rumors started to circulate: that the disease was not real, it was a ruse for political or monetary gain, that foreigners at the health centres were stealing body parts, that they were sorcerers that spread illness through magical means, that the foreigners were involved in genocidal plot to depopulate the area so that the land could be given to mining operations (ibid). As such, it is important to note that there exists a poignant cultural logic based on direct experiences that undergird such understandings and directly influence perceptions of risk.

## 4. Results and Discussion

### 4.1. Risk Perception and Behavioural Change

In light of the above discussion, we would expect that foreign workers’ claims about EVD were likely to be met with skepticism, and that their input would not likely lead to a change in local forms of risk perception. The analysis of our qualitative data reveals instead that what did contribute to changes in risk perception was the experience of seeing first-hand loved ones and members of the community succumb to the disease. One respondent from Lofa County said:


*“There were two stages; stage one was the period where people were dying. You know, [we thought] Ebola was not real, that in fact it was an invented virus, the white people wanted to kill us. Stage two was the period where government brought restrictions, regulations, markets were suspended, schools were suspended, in fact in 2014, the midterm election was extended.”*


Below we will return to this point regarding how such awareness experiences were transformative. For now, however, we will discuss how we may identify changes in risk perception. One indication that may demonstrate a change in risk perception is behavioral change—especially the adoption of, or receptiveness to, taking actions to avoid or prevent risk exposure. It logically follows that when the risk of EVD is perceived as real, there will be an increased willingness to adopt practices aimed at avoiding risks. We see this illustrated in several ways, but the avoidance of others is the most obvious example. One Lofa County respondent recounts:


*“When we heard about the symptoms of Ebola, headache, red eyes, and all kinds of symptoms they told us about. There came a time when I was feeling a headache so I distanced, I never disclosed it to my family, but I told them that everyone should be by themselves. I didn’t even play with my children. I was lonely and did that for almost 21 days but I never informed anyone about it.”*


In a similar light, another respondent from Lofa County recounts:


*“Many days we couldn’t go anywhere, even to the market. In fact, there was no market because everybody was afraid to go in the market. However, the hand washing stations that came, and the help that came, helped us avoid that sickness and allowed us to remain alive, but Ebola really disturbed plenty of things in Solomba.”*


Acceptance of the Ebola risk after the introduction of community-based interventions also led to a reconsideration of Ebola Treatment Units (ETUs) as places of threat. A respondent from Needowein, Margibi reflects on what he would do if he had gotten sick during the EVD outbreak:


*“Well, though I did not get sick, but if for any reason I was going to get sick, I would have preferred a specific ETU by then. And after that, when Ebola subsided, I was the only first person that embraced the Ebola survivor that came from the ETU and came into our community. I was the first person. So, because of me other people started coming closer to the survivors. Other people were looking at me interacting with the survivors and caring for them, and even though I interacted with them, they did see that I never contracted Ebola. And because of me, the people came closer to the survivors and never did bad things to them.”*


The community-based intervention also led to the adoption of greater distancing protocols. As one Nyanford Town resident noted:


*“My community did well in fighting Ebola. The community organized itself. Let me say that the community established a system to ensure that everybody could stay at home. They stopped all social activities and made sure that every family kept their children home.”*


### 4.2. Mobility

Physical distancing was also related to changes in mobility as people would consciously separate themselves in order to ensure non-contact with those potentially affected. This impacted on where, how and why people travelled during the outbreaks. The Lofa County respondent who spoke about the market closures above also recounts the restrictions placed on travelling in and out of the communities:


*“Yes, during the time Ebola was in our town, we would not allow people from other communities to come into our town. Why? Because we were afraid that somebody leaving from another community would come into our town and bring the virus with them. So even in the town itself, people were forbidden to leave their houses to visit their neighbors’ houses because nobody knew the status of their friends or family member. So, everybody was at his own house, and we could not just embrace strangers from outside in our community for fear of getting the disease. *There were also people trying to leave to stay safe elsewhere. Some people say that* Ebola was killing people here in Foya, so let me try to go to this [other] community. And most of the people, if you could, would try. However, if you see somebody travelling, be careful! Either that person is a contact, or he is sick and wants to move from one community to the other community.”*


In the Banjor/Red Hill community, another respondent recalls that:


*“Taxi drivers were afraid to even carry people because by then, four people weren’t allowed to sit together in the taxi and two people weren’t allowed to ride together on the motorbike.”*


### 4.3. Trust in Officials

Bearing in mind our discussion above and the effects of the colonial legacy on public trust, we do see that as the outbreak unfolded there was a discernable change in how community members viewed the various formal and informal officials involved in the response. This change is also seen with respect to the level of trust people invested in the various authorities. It was clear that during the early stages of the outbreak views of government authorities involved in the response were largely negative, as indicated for instance in the espousal of various conspiracy theories, suspicions of Ebola Treatment Units and hospitals, and criticism of the behavior of responders during the early response phases. This respondent from Needowein, Margibi remembers the initial responders in a particular way:


*“When the response team came, they would dress like beggars. They would not talk to you nicely, they will talk to you harshly. They come, spray the whole place [with chlorine] and go away. They would not talk to you politely. They were not giving us any encouragement. The only thing they wanted was to come and carry the person away so that they could get their money.”*


Another respondent from Needowein, Margibi notes a difference in his view of the response team after the adoption of the community-based protocol:


*“Yeah, while it is true that the response team was often delayed, the good thing about it was that at the end of the day, they did respond to the sick patients at various hospitals, at the clinic and at various homes. So, at the end of the day, they did what they could do in order to get the sick patient out of the various homes to the to the ETU center or the various hospitals.”*


Similarly, in Banjor/Red Hill, one respondent also recalls a transition:


*“Before the response team made us feel bad. However, later, they were trained to the extent that they could actually inform us properly. I think they were slow before because they lacked proper health worker training but after being trained [by the community-based initiative], they then came to the community and responded to our questions.”*


This respondent, also from Banjor/Red Hill, further outlines the shift in risk perception that encouraged communities to follow the guidelines:


*“To eradicate Ebola we realized we had to stick to the rules. We were willing to learn. There were those who didn’t believe that, they said Ebola was a money-making thing. However, eventually they let that belief go away. People were willing to stick to the rule of washing hands, stop hugging one another, stop kissing one another and then they were willing to go to the ETU whenever they fell sick, whenever they came down with the virus. They were willing to go because everything thing else wasn’t working.”*


### 4.4. Risk Information

Peoples’ views about authorities also influenced how community members processed the information they received, which in turn influenced their risk perceptions. Notably, how information was received and processed changed once community leaders got involved. These respondents from Needowein, Margibi demonstrate the changes in perception due to the information provided. One respondent did not trust the Ebola Treatment Units (ETUs):


*“If I became sick I wouldn’t have reported myself to the ETU because if you listened to the radio there was no word of encouragement. Nothing that said that if you go to the ETU you will come back alive. So, for me, if I became sick I was going to separate myself from my children and be alone.”*


Another recounts a shift once the information was the same from all authority groups:


*“Yeah, what I felt good about with the Ebola response was when all the different groups started to have the same specific message once the community leadership became involved in the response. Thank you.”*


This respondent affirms that information and messaging made a difference in eradicating EVD:


*“Like the others have been saying, we were the first group to start going around and talking to people, telling them about the virus, about washing your hands, staying away from people, staying away from gatherings. So that was our help. After that, the national government came in and gave the same message that we were giving. That they came in with the same message I think greatly helped in eradicating the virus.”*


### 4.5. Distrust of Institutions

As we have already seen, one indication of a lack of trust in official responders was the circulation of numerous conspiracy theories concerning how officials were benefiting from the outbreak. This led to the avoidance of Ebola Treatment Units and voicing of various suspicions such as the idea that Ebola was a threat deliberately fabricated to attract funds from outside the country [6]. Suspicion and distrust of officials was also expressed with respect to other aspects of the response, such as treatment in ETUs and the use of chlorine—which some believed was a poison used to harm people. A respondent from Lofa County describes the impact of mistrust of official responders:


*“They will say that the food and water in the ETU when you eat it or drink you will die. However, if we knew someone who went to ETU and survived, they will go to them and advise them. Most of them knew me and give me the opportunity to talk to them and some of them are alive today because of that. Most of our people died because of denial. Those that give us the opportunity to talk to them today they survived.”*


Another respondent, from Banjor/Red Hill, recounts the common misconceptions people held:


*“The information was that when the ambulance team came for you, they would carry you to the ETU and you will not live, you will not survive, you will die so they go and sprayed chorine on you and put chlorine in the food and give it to you, so even when somebody coming down with the virus when they here that information, they will not be in the position to go to report themselves they will run away.”*


Distrust in official responders was also bolstered by experiences with what was perceived to be poor government management of the outbreak in various respects. Two respondents from Banjor/Red Hill describe the lack of trust in government response to the EVD outbreak:


*“Definitely the response was slow maybe because of logistic problems. And you know it was the first time Ebola came to Liberia so people did not understand and did not know what to do. The government did not put into place what should have been the correct response or what should have been the real things. Because of logistics and other things and people were not being trained to properly monitor those effects and that definitely led to a slow response but later on, you know, at least it started to go on correctly.”*


Another respondent echoes this sentiment:


*“From the beginning the response team was very slow, but for me what made me feel more bad, you know, was the lack of government preparation, you know the ministry of health was not prepared for such a disease. The government did not have proper mechanisms to be able to respond to the citizens.”*


As NGOs became increasingly involved in the response there was also an accompanying shift in trust levels. As the situation worsened and people realized that new measures were required, they may have become more open to foreign (white) NGO involvement. A respondent from Lofa County recounts:


*“For about two months no one would come to our town because they believed they would get sick. This was a great cause of embarrassment for us. When we went to another town to buy salt we wouldn’t reveal where we were from. The white people came to us and said that the way people in the community are dying, to help stop the spread of the virus would you agree for us [i.e., the NGO] to bury your dead in your town instead of carrying them to Foya for burial? The chiefs called a meeting and said we should agree to this.”*


In Needowein, Margibi, this respondent says:


*“At the start everything was very bad. However, after the white people came everything was okay for us and they cared for the sick people first. I was worried about caring for my son, it was not easy but then the white people came and they took care of my little son and they told me not to worry. That everything will be all right. Once they said that would happen, I was so happy.”*


One consequence of the distrust in official responders was that some sought the services of traditional healers who were perceived as more trusted helpers. A respondent in Lofa County describes how:


*“During Ebola of course, hospitals were almost shut down. However, still in our community we had people who used to sell tablets [medication] from their bags. So secretly some used to go to those people and buy drugs from them and take them. That was helping us.”*


### 4.6. Community Tensions and Trust Building

Interpersonal conflicts within the community were especially problematic during the earlier stages of the outbreak response when institutional trust in official response organizations was found to be lacking. Indeed, in some cases, the mere association or involvement with such organizations resulted in stigmatization and thus involved a great deal of personal sacrifice for those bold enough to be associated with those organizations. This stigmatization was seen across the focus group locations. In Lofa County, one respondent who got involved had personal conflict remarks that:


*“I worked with government officials and my boss asked me to be a member of the community task force. The first thing I did was to encourage members of the community to construct the ETU. I was involved with the building of hand-washing stations and providing buckets for community members to wash their hands. I was also involved in the meeting where they provided the land for the cemetery. So, most of the time that I was with them, I was creating awareness for people. However, in the end, I could not go home because my people said they bribed me. In fact, they gave me a big phone. My own wife and children rejected me, so I left the village for some time.”*


In Needowein, Margibi, a respondent describes the community divisions:


*“Yeah with Ebola coming into the community, it brought division within communities, family and let say neighbors. Because, for instance if you saw your neighbor sick, the rules said we had report that and then the response team would come and care for that person. The rest of the family of that person will be angry with you. People were no longer really free with one another and this scattered them because some people will not interact with each other and avoid each other. […]”*


By and large though, community-based initiatives did encourage better trust relations that led to better practices in various ways. In Banjor/Red Hill, this respondent recalls:


*“Liberians realized, especially the highly hit communities, that when they listened to the community health workers, they started to improve. People started going to the ETU and they started coming back alive. People came back with a success story, and they went to the media and explained that. This information started to spread so our approach was far better so immediately.”*


Finally, in West Point, this respondent recounts a rebuilding of trust when he perceived that the information was correct, and he began to see others in the community recover:


*“The trust came about because of the information. You know when Ebola started, information was not coming directly, but we start to get the right information. Following the directives, people went to the ETU and came back alive. You know, this brought the trust and confirmed to others that we were talking to in the field that we were telling them was based on true information.”*


Overall, this study is subject to the usual limitations associated with qualitative data analyses. The most prominent of these relate to issues of generalizability. In contrast to quantitatively oriented studies based on random sampling for statistical representativeness, this study was based on purposive sampling in which efforts are made to select a sample adequate to the task of theoretical explication. Thus, the methodological emphasis is not to generalize to the population, but to gain insight into the subjective dimensions of the phenomena under investigation—in this case, to gain a deeper understanding of the social context in which risk perceptions and relations of trust operate within the setting of the EVD outbreak response in West Africa.

## 5. Conclusions

The Ebola Virus Disease (EVD) outbreak response in Liberia in 2014–2015 was challenging because of an initial resistance of communities to adopt safety and prevention measures outlined by officials. This resistance was grounded in a history of mistrust of government, private extraction industries, and NGOs that were corrupt, exploitative, and mistreated locals. This pre-existing atmosphere of mistrust fueled the spread of misinformation about EVD and influenced people’s risk perception surrounding the virus. In light of an ineffectual response, community-based initiatives were introduced. These were based on a strategy of employing trusted local community leaders to help educate and encourage behavioral changes. The positive effects of the community-based initiatives were seen in terms of: the adoption of preventive practices such as self-isolation, the self-limitation of travel, greater hand-washing, and the increased likelihood of going to an Ebola Treatment Unit (ETU) if you had symptoms. On the basis of our findings, we conclude that when examining infectious disease response, local contexts alongside necessary medical interventions should be given due consideration. Secondly, community-based initiatives offer a more integrated epidemiological response for better community health outcomes. Thus, one important practical implication of this study is that outbreak responders, in addition to their medical and health training, should receive explicit and more contextually-specific training concerning the local politics and history of the region in which they are conducting their important work. This will facilitate better working relations between community outsiders and those residing within the particular community affected by the disease outbreak.

## Figures and Tables

**Figure 1 ijerph-18-03270-f001:**
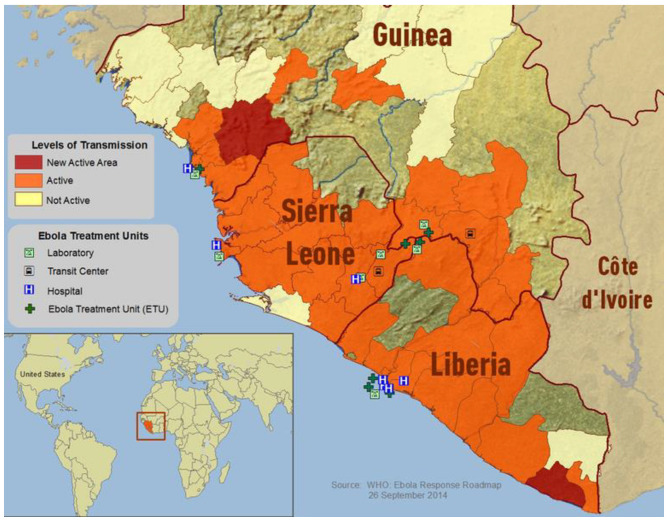
WHO Ebola 2014 outbreak map of Guinea, Liberia, and Sierra Leone. Source: Retreived from https://commons.wikimedia.org/wiki/File:Ebola_2014_outbreak_map_of_Guinea,_Liberia,_and_Sierra_Leone.png (accessed on 10 March 2021).

**Figure 2 ijerph-18-03270-f002:**
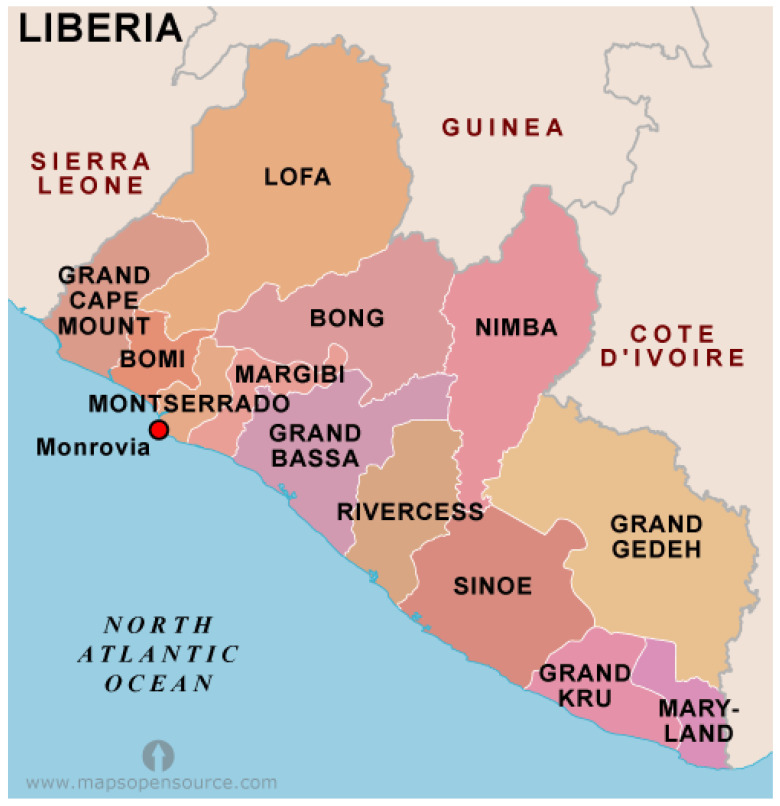
Counties of Liberia. Source: Retreived from http://www.mapsopensource.com/liberia-counties-map.html (accessed on 10 March 2021).

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
