# Peer review of "Contextualizing Risk Perception and Trust in the Community-Based Response to Ebola Virus Disease in Liberia"

_ijerph, 2021, doi:10.3390/ijerph18063270_

Round 1

Reviewer 1 Report

Overall I find this to be an excellent paper; indeed one of the best I have reviewed for this journal. It is well written, compelling, and addresses an important issue (community responses in times of crisis as a key driver of change), one that is flagged in much of the disaster response literature (which, incidentally, might be a more general framing the authors might consider in terms of broader implications). The methods are appropriate and well described. And the topic is well contextualized in terms of local socio-political-economic history. With a bit of tweaking it really should be required reading for all public health and global health training programs.

One aspect that doesn't come through clearly enough is what the community-based response and involvement actually was. We get glimpses of it through the extensive quotes (which are excellent, by the way), but it would be helpful to have an added paragraph that spells it out more clearly.

A few additional suggestions:

  1. One typo: "ours study" should be "our study" (p.2, line 95)
  2. Map: could show location of Liberia in Africa as an inset
  3. Given the range of types of respondents/participants included in the study, it would have been helpful to know what kind of participant was being quoted with each quote, if this could be done without compromising confidentiality

Author Response

Please see the attached document addressing reviewer comments

Reviewer 2 Report

The authors should discuss the acquired datasets.

What methodology is followed to conduct the proposed work. A flowchart should be presented.

The Paper looks more theoretical.

Author Response

(The authors gave the same response as above.)

Reviewer 3 Report

Thank you very much for being able to review this manuscript. It seems to me an interesting research, a subject little explored. For its publication, I would advise you to make some modifications:
Abstract: I recommend defining clearly what is the objective of the study, describe methodology, participants. Results should be clarified in the abstract.
In the methodology section there should be a table with the characteristics of the participants, and who participated in the different techniques. Inclusion criteria of the participants should be specified. What questions were asked to the participants? How was the analysis of the speeches carried out?
In results and discussion section, there is hardly any discussion of these results.
Limitations of the study should appear at the end of the results. Are there future lines of research? 
What are the implications of this study? The following should appear

Author Response

(The authors gave the same response as above.)

Reviewer 4 Report

The authors use qualitative data to contextualize’ Risk Perception and Trust in the Community-Based Response to Ebola Virus Disease in Liberia’.  The authors assert a need for this study by stating that the role of community-based initiatives in outbreak response is often overlooked in traditional epidemiological accounts.

 In lines 16-17, authors state ‘Using qualitative interview data from five distinct communities in Liberia we discuss how such shifts occurred”.  This sentence is unclear.  The authors should clarify what shifts occurred and among whom.  Or, this sentence may be better placed after the sentence in lines 17-19. 

There is an insightful definition of community-based initiatives in line 68-70.

In the Materials and Methods section, the recruitment of participants and attention to ethical considerations is well described.

The study is bolstered by a solid sample size of 100 participants from five distinct Liberian communities’ sites, including focus groups consisting of ten participants, and ten key informant interviews.

It is a bit difficult to follow when the authors go from describing the communities elected for the study starting in line 122 to discussing data analyses and study limitations in the next paragraph.  Perhaps the next step before discussing the analyses would be to discuss the interview questions.  Was an interview guide used?  What questions were participants asked?

It would be helpful to know more about the demographic characteristics of study participants, as this could add context to the qualitative responses reported in the paper.  Was this information collected?

Having the results and discussion grouped together is a bit confusing.  Perhaps the results should be presented first.  The study results should also be presented more clearly and cogently.

Overall, this is an interesting and insightful study.  There are a few clarifying questions, including about the intervention design, that the authors can explain.  The methodology needs to be clarified.  The authors should also consider reorganizing sections of the paper to help improve paper flow.  Attending to these items may help to improve the paper.

Author Response

(The authors gave the same response as above.)

Round 2

Reviewer 2 Report

Authors have improved the paper by adding the suggested comments, If other reviewers agree paper can go for publications.

Reviewer 3 Report

Suggested changes have been made. Manuscript can be accepted.

Reviewer 4 Report

In line 23 in abstract- Even though it is assumed that NGO is an abbreviation for non-governmental organization, it would be good to spell out the acronym the first time that it is introduced, for example 'non-governmental organization (NGO)'.

Otherwise, the authors seem to have sufficiently addressed reviewer feedback.  The paper is clearer and should be suitable for publication.